# A Simple Unified Method for Node Classification on Homophilous and Heterophilous Graphs

## Abstract

In graph learning, there have been two predominant inductive biases regarding graph-inspired architectures: On the one hand, higher-order interactions and message passing work well on homophilous graphs and are leveraged by GCNs and GATs. Such architectures, however, cannot easily scale to large real-world graphs. On the other hand, shallow (or node-level) models using ego features and adjacency embeddings work well in heterophilous graphs. In this work, we propose a novel scalable shallow method – GLINKX – that can work both on homophilous and heterophilous graphs. GLINKX leverages (i) novel monophilous label propagations (ii) ego/node features, (iii) knowledge graph embeddings as positional embeddings, (iv) node-level training, and (v) low-dimensional message passing. Formally, we prove novel error bounds and justify the components of GLINKX. Experimentally, we show its effectiveness on several homophilous and heterophilous datasets with up to millions of nodes and tens of millions of edges.

## 1 Introduction

In recent years, graph learning methods have emerged with a strong performance for various ML tasks. Graph ML methods leverage the topology of graphs underlying the data (Battaglia et al., 2018) to improve their performance. Two very important design options for proposing graph ML-based architectures in the context of *node classification* are related to whether the data is *homophilous* or *heterophilous*.

For homophilous data – where neighboring nodes share similar labels (McPherson et al., 2001; Altenburger & Ugander, 2018a) – Graph Neural Network (GNN)-based methods can achieve high accuracy. Specifically, a broad subclass of successful GNNs are Graph Convolutional Networks (GCNs) (e.g., GCN, GAT, etc.) (Kipf & Welling, 2016; Veličković et al., 2017; Zhu et al., 2020). In the GCN paradigm, *message passing* and *higher-order interactions* help node classification tasks in the homophilous setting since such inductive biases tend to bring the learned representations of linked nodes close to each other. However, GCN-based architectures suffer from *scalability issues*. Performing (higher-order) propagations during the training stage are hard to scale in large graphs because the number of nodes grows exponentially with the increase of the filter receptive field. Thus, for practical purposes, GCN-based methods require *node sampling*, substantially increasing their training time. For this reason, architectures (Huang et al., 2020; Zhang et al., 2022b; Sun et al., 2021; Maurya et al., 2021; Rossi et al., 2020) that leverage propagations outside of the training loop (as a preprocessing step) have shown promising results in terms of scaling to large graphs since they do not require neighborhood sampling. Moreover, deploying GCN-based methods on an industrial scale incurs infrastructure overhead since training GNNs in the large-scale regime with node sampling requires significant computing resources (i.e. more GPU memory) compared to simpler shallow methods that perform i.i.d.-based random sampling.

In *heterophilous* datasets (Rogers et al., 2014), the connected nodes tend to have different labels. Currently, many works that address heterophily can be classified into two categories concerning scale. On the one hand, recent successful architectures (in terms of accuracy) (Rossi et al., 2023; Di Giovanni et al., 2022; Zheng et al., 2022b; Luan et al., 2021; Chien et al., 2020; Lei et al., 2022) that address heterophily resemble GCNs in terms of design and and therefore face the same scalability problems as GCNs/GATs. On the other hand,

*shallow or node-level models* (see, e.g., (Lim et al., 2021; Zhong et al., 2022)), i.e., models that are treating graph data as tabular/i.i.d. data, resemble simple multi-layer perceptrons (MLPs) in their architecture, and do not involve propagations during training, have shown a lot of promise for large heterophilous graphs. For instance, in Lim et al. (2021), it is shown that combining ego embeddings (node features) and adjacency embeddings has good accuracy and scalability in the heterophilous setting (see Section 2). However, their design is still impractical in real-world data since the LINKX method presented in their paper is *not* inductive (see Section 2), and embedding the adjacency matrix exactly requires many parameters in a model, which can be millions to billions of extra parameters for industry-scale networks. In LINKX, the adjacency embedding of a node can alternatively be thought of as a *positional embedding* (PE)[1] of the node in the graph, and recent developments (Kim et al., 2022; Dwivedi et al., 2021; Lim et al., 2021; Wang et al., 2022) have shown the importance of PEs in both homophilous and heterophilous settings. However, most of these works suggest PE parameterizations (e.g., Laplacian eigenvectors) that are difficult to compute in large-scale directed graphs, such as modern social networks. We argue that a way to circumvent this problem is to rely on *knowledge graph embeddings* (Bordes et al., 2013; Yang et al., 2014) which can be used to perform a non-linear factorization of the adjacency matrix of the network and have been recently shown to be able to be trained in billion-scale networks (El-Kishky et al., 2022; Lerer et al., 2019).

**Goal & Contribution**

We present GLINKX, a simple, scalable, and effective shallow method that works on both homophilous and heterophilous graphs that address the scalability problems of GNNs as well as the accuracy and memory inefficiency problems that shallow heterophilous methods face. For a method to be scalable, we argue that it should: *(i)* run models on node-scale (thus leveraging i.i.d. minibatching), *(ii)* avoid doing message passing during training and do it a constant number of times before training, and *(iii)* transmit small messages along the edges (to reduce memory footprint).

To develop GLINKX, our main structural intuition is that many real-world homophilous and heterophilous datasets exhibit the well-documented *monophily* property (see Section 2.5, and Altenburger & Ugander (2018a); Lim et al. (2021); Altenburger & Ugander (2018b)); namely, a node tends to associate with nodes that have the same class as one another (i.e., the node has unique types of "friends"). This property can hold regardless of the graph being homophilous or heterophilous (see Figures 1(b) and 1(c)).

Given this real-world intuition, GLINKX tackles the problems that current methods suited for homophily and heterophily face by combining three simple, novel, and effective components:

1. A novel 2-hop propagation scheme called *MLaP* (see Section 3 and Figure 3) which performs propagations outside of the training loop and addresses the infrastructure bottlenecks of the message-passing architectures, as well as the decreased performance of shallow models in homophilous graphs. MLaP relies on the structural assumption that many real-world homophilous and heterophilous graphs are monophilous.
2. Knowledge Graph Embeddings (KGEs) (see also Sections 2.6 and 3) to compress the adjacency graph representations that existing methods such as LINKX use and provide positional information (positional embeddings) about the nodes.
3. The ego embeddings of each node which have been shown to work in both the homophilous and heterophilous context[2].

We show that GLINKX can perform well on a variety of homophilous and heterophilous datasets ranging from a few thousand nodes and edges to millions and tens of millions of nodes and edges (Section 4). Moreover, we provide novel theoretical error bounds to justify the components of GLINKX (see Section 3.3). Even though the state-of-the-art methods (see, e.g., Luan et al. (2021); Rossi et al. (2023)), outperform our method in terms of accuracy; such methods are inherently not scalable to large-scale datasets because they perform propagations (message-passing) during training which make neighborhood sampling mandatory to be able to run on large datasets. In these cases, training GNNs on large datasets is still feasible; yet, it is very costly

---

[1]We use the word *"positional embedding"* to talk broadly about embeddings that correspond to the nodes of a graph and encode the structural characteristics of each node.

[2]We use ego embeddings and node features interchangeably.

in terms of time, cost, and resources and takes up considerable time compared to our method (cf. Frasca et al. (2020); Bojchevski et al. (2020) and the references therein, and the runtime comparison in Section 4). Our method overcomes this bottleneck by performing propagations twice and out of the training loop, which makes it easier to get deployed in industrial applications since our propagations can be implemented with modern distributed storage and processing software such as Apache Hadoop. Moreover, we also argue that GLINKX is complementary to what other methods propose, and such complementary information can be included in GLINKX without sacrificing performance. Finally, GLINKX is suitable for industrial applications since it can work in both a *transductive* and an *inductive*[3] setting.

## 2 Preliminaries

### 2.1 Notation

We denote scalars with lower-case, vectors with bold lower-case letters, and matrices with bold upper-case. We consider a directed graph $G = G(V, E)$ with vertex set $V$ with $|V| = n$ nodes, and edge set $E$ with $|E| = m$ edges, and adjacency matrix $\boldsymbol{A}$. $\boldsymbol{X} \in \mathbb{R}^{n \times d_X}$ represents the $d_X$-dimensional features and $\boldsymbol{P} \in \mathbb{R}^{n \times d_P}$ represent the $d_P$-dimensional PE matrix (see Section 2.6 and Appendix A.2). A node $i$ has a feature vector $\boldsymbol{x}_i \in \mathbb{R}^{d_X}$ and a positional embedding $\boldsymbol{p}_i \in \mathbb{R}^{d_P}$ and belongs to a class $y_i \in \{1, \ldots, c\}$. The training set is denoted by $G_{\text{train}}(V_{\text{train}}, E_{\text{train}})$, the validation set by $G_{\text{valid}}(V_{\text{valid}}, E_{\text{valid}})$, and test set by $G_{\text{test}}(V_{\text{test}}, E_{\text{test}})$. $\mathbb{I}\{\cdot\}$ is the indicator function. $\Gamma_c$ is the $c$-dimensional simplex.

### 2.2 Graph Convolutional Neural Networks

In homophilous datasets, GCN-based methods have been used for node classification. GCNs (Kipf & Welling, 2016) utilize feature propagations together with non-linearities to produce node embeddings. Specifically, a GCN consists of multiple layers where each layer $i$ collects $i$-th hop information from the nodes through propagations and forwards this information to the $i + 1$-th layer. More specifically, if $G$ has a symmetrically-normalized adjacency matrix $\boldsymbol{A}'_{sym}$ (with self-loops) (ignoring the directionality of edges), then a GCN layer has the form

$$\boldsymbol{H}^{(0)} = \boldsymbol{X}, \boldsymbol{H}^{(i+1)} = \sigma\left(\boldsymbol{A}'_{sym}\boldsymbol{H}^{(i)}\boldsymbol{W}^{(i)}\right) \quad \forall i \in \{1, \ldots, L\},$$

$$\boldsymbol{Y} = \text{softmax}\left(\boldsymbol{H}^{(L)}\right).$$

Here $\boldsymbol{H}^{(i)}$ is the embedding from the previous layer, $\boldsymbol{W}^{(i)}$ is a learnable projection matrix and $\sigma(\cdot)$ is a non-linearity (e.g. ReLU, sigmoid, etc.).

### 2.3 Message-Passing Architectures vs. Efficient Shallow Methods

***Message-Passing Architectures:*** To train a GCN-based model (or generally, whenever message-passing is involved) on a large network (that cannot fit in the GPU memory), one has to do *minibatching* through *neighbor sampling*. For large-scale networks, mini-batching takes much longer than full-batch training and requires substantially more infrastructure, which is one of the reasons that graph GCNs are not preferred in real-world settings (see, e.g., Jin et al. (2022b); Zhang et al. (2022a); Zheng et al. (2022a); Lim et al. (2021); Maurya et al. (2021); Rossi et al. (2020)).

***Efficient Shallow Methods:*** *Shallow* (or node-level) models are based on manipulating the node features $\boldsymbol{X}$ and the graph topology $\boldsymbol{A}$ so that propagations do not occur during training. Such methods treat the input embeddings as tabular data and pass them through a feed-forward neural network (MLP) to produce the predictions. Thus, they avoid the message-passing bottlenecks and instead rely on simple tabular mini-batching. For this reason, most methods that can scale on real-world settings are *shallow*. In heterophilous

---

[3]For this paper, we operate in the *transductive setting*. See Appendix B for the inductive setting.

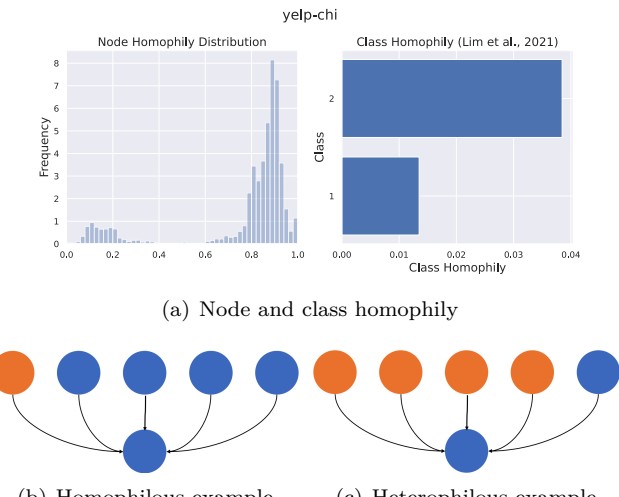

(a) Node and class homophily

(b) Homophilous example     (c) Heterophilous example

Figure 1: Top: Node and class homophily distributions for the yelp-chi dataset. Bottom: Examples of a homophilous (Figure 1(b)) and a heterophilous (Figure 1(c)) region in the same graph that are both monophilous, namely they are connected to many neighbors of the same kind. In a spam network, the homophilous region corresponds to many non-spam reviews connecting to non-spam reviews (which is the expected behavior of a non-spammer user). The heterophilous region corresponds to spam reviews targeting non-spam reviews (the expected behavior of spammers), thus, yielding a graph with both homophilous and heterophilous regions such as in Figure 1(a).

datasets, the simple method of LINKX has been shown to perform well. LINKX combines two components – MLP on the node features $\boldsymbol{X}$ and LINK regression (Altenburger & Ugander, 2018a) on the adjacency matrix – as follows:

$$\boldsymbol{H}_X = \mathrm{MLP}_X(\boldsymbol{X})\,,\ \boldsymbol{H}_A = \mathrm{MLP}_A(\boldsymbol{A})\,,\ \boldsymbol{Y} = \mathrm{ResNet}(\boldsymbol{H}_X, \boldsymbol{H}_A).$$

Examples of other methods include FSGNN (Maurya et al., 2021), and SIGN (Rossi et al., 2020).

## 2.4 Node Classification

In graph node classification, we have a model $f(\boldsymbol{X}, \boldsymbol{Y}_{\mathrm{train}}, \boldsymbol{A}; \boldsymbol{\theta})$ that takes as an input the node features $\boldsymbol{X}$, the training labels $\boldsymbol{Y}_{\mathrm{train}}$ and the graph topology $\boldsymbol{A}$ and produces a prediction for each node $i$ of $G$, which corresponds to the probability that a given node belongs to any of $c$ classes (with the sum of such probabilities being one). The model is trained with back-propagation. Once trained, the model can be used for the prediction of labels of nodes in the test set.

There are two training regimes: *transductive* and *inductive*. In the transductive training regime, we have full knowledge of the graph topology (for the train, test, and validation sets) and the node features, and the task is to predict the labels of the validation and test set. In the inductive regime, only the graph induced by $V_{\mathrm{train}}$ is known at the time of training, and then the full graph is revealed for prediction on the validation and test sets. In real-world scenarios, such as online social networks, the dynamic nature of problems makes the inductive regime particularly useful.

## 2.5 Homophily, Heterophily & Monophily

***Homophily and Heterophily:*** There are various measures of homophily in the GNN literature like node homophily and edge homophily (Lim et al., 2021). Intuitively, homophily in a graph implies that nodes with similar labels are connected. GNN-based approaches like GCN, GAT, etc., leverage this property to improve the node classification performance. Alternatively, if a graph has low homophily – namely, nodes that

connect tend to have different labels – it is said to be *heterophilous*. In other words, a graph is heterophilous if neighboring nodes do not share similar labels.

***Monophily:*** Generally, we define a graph to be monophilous if the label of a node is similar to that of its neighbors' neighbors[4]. Etymologically, the word "monophily" is derived from the Greek words *"monos"* (unique) and *"philos"* (friend), which in our context means that a node – regardless of its label – has neighbors of primarily one label. In the context of a directed graph, monophily can be thought of as a structure that resembles Figure 3(a) where similar nodes (in this case, three green nodes connected to a yellow node) are connected to a node of different/same label.

## 2.6 Knowledge Graph Embeddings as Positional Embeddings

***Knowledge Graphs:*** Knowledge graph embeddings are a way to present knowledge about the world in a structured way. They consist of triplets $(h, r, t)$, which correspond to a head $(h)$, a relation $(r)$, and a tail $(t)$, such that the tail $t$ is related to the head $h$ via the relation $r$. The union of all such triplets defines a heterogeneous graph, the Knowledge Graph $G(V, E_1, \ldots, E_R)$ where the number of relations is $R$. Knowledge graphs can have multiple relations that represent different associations of objects, for example, $(\mathsf{Paris}, \mathsf{isCapitalOf}, \mathsf{France})$, and $(\mathsf{Baguette}, \mathsf{isEatenIn}, \mathsf{France})$.

***Knowledge Graph Embeddings:*** The aim of knowledge graph embeddings (KGEs) is to provide continuous-space representations for the entities $\{\boldsymbol{p}_u\}_{i \in V} \mathbb{R}^{|V| \times d_P}$ and the relations $\{\boldsymbol{r}_l\}_{l \in [R]} \in \mathbb{R}^{R \times d_P}$, such that for a triplet $(\boldsymbol{h} = \boldsymbol{p}_u, \boldsymbol{r} = \boldsymbol{r}_l, \boldsymbol{t} = \boldsymbol{p}_v)$, $\boldsymbol{h} + \boldsymbol{r} \approx \boldsymbol{t}$ where "$\approx$" corresponds to minimizing a distance criterion. Training is done by sampling negative examples for each positive example $(\boldsymbol{h}, \boldsymbol{r}, \boldsymbol{t})$ and minimizing a contrastive-type loss. There have been numerous methods proposed for modeling and training KGEs, see Wang et al. (2017) for an in-depth literature review. In our paper, we use the DistMult method introduced by Yang et al. (2014) where the distance criterion is the triple Hadamard (element-wise) product $\boldsymbol{h} \odot \boldsymbol{r} \odot \boldsymbol{t}$.

Moreover, we note that *homogeneous* graphs – namely graphs where there is only one relation (for example the simple edge relation) – are a subset of knowledge graphs where all nodes and edges have the same types. In this paper, we use KGEs on homogeneous graphs as a way to extract embeddings for the nodes of the graph.

***KGEs as Positional Embeddings:*** Apart from representing knowledge in a continuous space, KGEs can provide positional information – i.e., positional embeddings – for the graph nodes. Specifically, training knowledge graph embeddings can be seen as performing **a non-linear factorization on the adjacency matrices** in order to generate embeddings for the nodes. These embeddings can then be used as positional embeddings (PEs) for representing the "position of a node" in the graph (see Section 3), which serves as a useful additional signal for node classification in both homophilous and heterophilous graphs (cf. Lim et al. (2021); Dwivedi et al. (2021); Srinivasan & Ribeiro (2019); Wang et al. (2022)). Finally, KGEs can be trained scalably for graphs with billions of nodes (see, e.g., El-Kishky et al. (2022)). For our paper, we describe the KGE training method in Appendix A.2.

# 3 Method

## 3.1 Components & Motivation

The desiderata we laid down on Section 1 can be realized by three components: (i) PEs, (ii) ego embeddings, and (iii) label propagations that encode monophily. More specifically, ego embeddings and PEs are used as primary features, which have been shown to work for both homophilous and heterophilous graphs for the models we end up training. Finally, the propagation step is used to encode monophily to provide additional information to our final prediction.

***Positional Embeddings:*** We use PEs to provide our model information about the position of each node and hypothesize that PEs are an important piece of information in the context of large-scale node classification.

---

[4]A similar definition of monophily has appeared in (Altenburger & Ugander, 2018a), whereby many nodes have extreme preferences for connecting to a certain class.

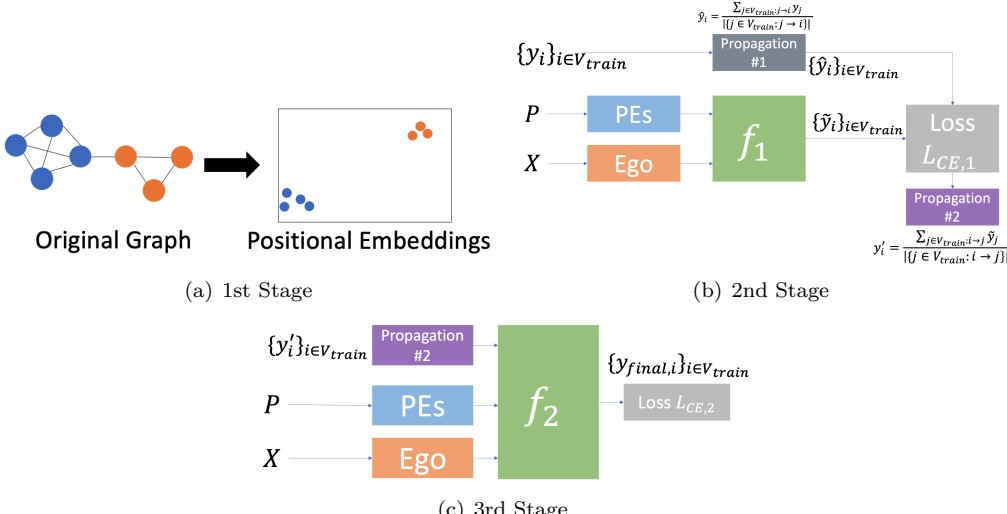

Figure 2: Block Diagrams of GLINKX stages.

PEs have been used to help discriminate isomorphic graph (sub)-structures (Kim et al., 2022; Dwivedi et al., 2021; Srinivasan & Ribeiro, 2019), and also architectures that are able to learn node representations jointly and PEs have been developed (Wang et al., 2022). Specifically, Wang et al. (2022) develops a method to learn PEs during GNN training by using a separate channel to update the original node features and the PEs. Their architecture (PEG Layer) is permutation-invariant wrt the node features, rotationally and reflectively invariant wrt the PEs, and has good stability guarantees. We note that the difference with our method is that in our method the PEs are pre-trained and provided externally, rather than trained in an end-to-end manner.

PEs are useful for both homophily (Kim et al., 2022; Dwivedi et al., 2021) and heterophily (Lim et al., 2021) because isomorphic (sub)-structures can exist in both the settings. In the homophilous case, adding positional information can help distinguish nodes that have the same neighborhood but distinct position (Dwivedi et al., 2021; Morris et al., 2019; Xu et al., 2019), circumventing the need to do higher-order propagations (Dwivedi et al., 2021; Li et al., 2019; Bresson & Laurent, 2017) which are prone to over-squashing (Alon & Yahav, 2021). In heterophily, structural similarity among nodes is important for classification, as in the case of LINKX – where adjacency embedding can be considered a PE. However, in large graphs, using adjacency embeddings or Laplacian eigenvectors be a computational bottleneck and may be infeasible (cf. Kim et al. (2022)).

In this work, we leverage *knowledge graph embeddings* (KGEs) to encode positional information about the nodes, and embed the graph, as a way to perform *a non-linear factorization on the adjacency matrix of the graph.* The resulting factorization can serve as a graph embedding – as we describe in Section 2.6 – which can be utilized as a PE. For our paper, we consider a simple case of the graph having *only* one relation – also called a *homogeneous graph* – which represents the topological links in the graph (i.e., edges).

Using KGEs has two benefits: Firstly, KGEs can be trained easily and efficiently in many real-world scenarios (cf. El-Kishky et al. (2022)). This is because KGEs compress the adjacency matrix into a fixed-sized embedding. Further, KGEs are lower-dimensional than the adjacency matrix (e.g., $d_P \sim 10^2$), allowing for faster training and inference times, as well as better utilization of machine learning infrastructure. Secondly, KGEs can be pre-trained efficiently on such graphs (Lerer et al., 2019) and can be used off-the-shelf for other downstream tasks, including node classification (El-Kishky et al., 2022)[5]. So, in the 1st Stage of GLINKX in Algorithm 1 (Figure 2(a)) we train KGEs model on the available graph structure. Here, we fix this positional encoding once they are pre-trained for downstream usage. Finally, we note that this step is transductive but we can easily make it inductive (El-Kishky et al., 2022; Albooyeh et al., 2020).

---

[5]Positional information can also be provided by other methods such as node2vec (Grover & Leskovec, 2016) or LINE (Tang et al., 2015), however, most of such methods are less scalable.

***Ego Embeddings:*** We get ego embeddings from the node features. Such embeddings have been used in homophilous and heterophilous settings (Lim et al., 2021; Zhu et al., 2020). Node embeddings are useful for tasks where the graph structure provides little/no information about the task.

***Monophilous Label Propagations:*** We now propose a novel monophily (refer Section 2.5) inspired label propagation which we refer to as Monophilous Label Propagation (MLaP). MLaP has the advantage that we can use it both for homophilous and heterophilous graphs or in a scenario with varying levels of graph homophily (see, e.g., Figure 1).

*Why Encode Monophily?* We argue that encoding monophily into a model *can be helpful for both heterophilous and homophilous graphs (see Figures 1(b) and 1(c)), which is one of the main motivators behind our work.* In homophilous graphs, monophily will fundamentally encode the 2nd-hop neighbor's label information, and since in such graphs, neighboring nodes have similar labels, it can provide a helpful signal for node classification. In heterophily, neighboring nodes have different labels, but the 2nd-hop neighbors may share the same label, providing helpful information for node classification. Monophily is effective for heterophilous graphs (Lim et al., 2021). Therefore, an approach encoding monophily has an advantage over methods designed specifically for homophilous and heterophilous graphs, especially when varying levels of homophily can exist between different sub-regions in the same graph (see Figure 1). It may also not be apparent if the (sub-)graph is purely homophilous/heterophilous (since these are not binary constructs), which makes a unified architecture that can leverage graph information for both settings all the more important.

*How does MLaP encode monophily?* To understand how MLaP encodes monophily, we consider the example in Figure 3. In this example, we have three green nodes connected to a yellow node and two nodes of different colors connected to the yellow node. Then, one way to encode monophily in Figure 3(a) while predicting label for $j_\ell, \ell \in [5]$, is to get a *distribution* of labels of nodes connected to node $i$ thus encoding its neighbors' distribution. The fact that there are more nodes with green color than other colors can be used by the model to make a prediction. But this information may only sometimes be present, or there may be few labeled nodes around node $i$. Consequently, we propose a 2-step approach to get this information. First, we train a model that predicts the label distribution of nodes connected to $i$. We use the node features ($\boldsymbol{x}_i$) and PE ($\boldsymbol{p}_i$) of node $i$ to build this model since nodes that are connected to node $i$ share similar labels and thus, the features of node $i$ must be predictive of its neighbors. So, in Figure 3(a), we train a model to predict a distribution of $i$'s neighbors. Next, we provide $j_\ell$ the learned distribution of $i$'s neighbors by propagating the learned distribution from $i$ back to $j_\ell$, and therefore now $j_\ell$ has information about $i$'s neighbors. Equations (1) to (3) correspond to MLaP. We train a final model that leverages this information together with node features and PEs (Figure 3(b)).

## 3.2   Our Method: GLINKX

We put the components discussed in Section 3.1 together into three stages. In the first stage, we pre-train the PEs by using KGEs. Next, encode monophily into our model by training a model that predicts a node's neighbors' distribution and by propagating the soft labels from the fitted model. Finally, we combine the propagated information, node features, and PEs to train a final model. GLINKX is described in Algorithm 1 and consists of three main components detailed as block diagrams in Figure 2. Figure 3 shows the GLINKX stages from Algorithm 1 on a toy graph:

*1st Stage (KGEs):* We train DistMult KGEs with Pytorch-Biggraph (Yang et al., 2014) treating $G$ as a knowledge graph with only one relation (see Appendix A.2 for more details). Here, we have decided to use DistMult, but one can use their method of choice to embed the graph.

*2nd Stage (MLaP):* First (2nd Stage in Algorithm 1, Figure 2(b), and Figure 3(a)), for a node we want to learn the distribution of *its neighbors.* To achieve this, we propagate the labels from a node's neighbors (we call this step MLaP Forward), i.e., calculate

$$\hat{\boldsymbol{y}}_i = \frac{\sum_{j \in V_{\text{train}}:(j,i) \in E_{\text{train}}} \boldsymbol{y}_j}{|\{j \in V_{\text{train}} : (j,i) \in E_{\text{train}}\}|} \qquad \forall i \in V_{\text{train}}. \tag{1}$$

---

**Algorithm 1** GLINKX Algorithm

---

**Input:** Graph $G(V, E)$ with train set $V_{\text{train}} \subseteq V$, node features $\boldsymbol{X}$, labels $\boldsymbol{Y}$
**Output:** Node Label Predictions $\boldsymbol{Y}_{\text{final}}$
**1st Stage (KGEs).** Pre-train knowledge graph embeddings $\boldsymbol{P}$ with Pytorch Biggraph.
**2nd Stage (MLaP).** Propagate labels and predict the neighbor distribution

    1. **MLaP Forward:** Calculate the distribution of each training node's neighbors, i.e.
$\hat{\boldsymbol{y}}_i = \frac{\sum_{j \in V_{\text{train}}:(j,i) \in E_{\text{train}}} \boldsymbol{y}_j}{|\{j \in V_{\text{train}}:(j,i) \in E_{\text{train}}\}|}$ for all $i \in V_{\text{train}}$

    2. **Learn distribution of a node's neighbors:**

      (a) For each epoch, calculate $\tilde{\boldsymbol{y}}_i = f_1(\boldsymbol{x}_i, \boldsymbol{p}_i; \boldsymbol{\theta}_1)$ for $i \in V_{\text{train}}$

      (b) Update the parameters s.t. the negative cross-entropy $\mathcal{L}_{\text{CE},1}(\boldsymbol{\theta}_1) = \sum_{i \in V_{\text{train}}} \text{CE}(\hat{\boldsymbol{y}}_i, \tilde{\boldsymbol{y}}_i; \boldsymbol{\theta}_1)$ is maximized in order to bring $\tilde{\boldsymbol{y}}_i$ statistically close to $\hat{\boldsymbol{y}}_i$.

      (c) Let $\boldsymbol{\theta}_1^*$ be the parameters at the end of the training that correspond to the epoch with the best validation accuracy.

    3. **MLaP Backward:** Calculate $\boldsymbol{y}_i' = \frac{\sum_{j \in V:(i,j) \in E} \tilde{\boldsymbol{y}}_j}{|\{j \in V:(i,j) \in E\}|}$ for all $i \in V_{\text{train}}$, where $\tilde{\boldsymbol{y}}_j = f_1(\boldsymbol{x}_j, \boldsymbol{p}_j; \boldsymbol{\theta}_1^*)$.

**3rd Stage (Final Model).** Predicting node's own label distribution:

    1. For each epoch, calculate $y_{\text{final},i} = f_2(\boldsymbol{x}_i, \boldsymbol{p}_i, \boldsymbol{y}_i'; \boldsymbol{\theta}_2)$.

    2. Update the parameters s.t. the negative cross-entropy $\mathcal{L}_{\text{CE},2}(\boldsymbol{\theta}_2) = \sum_{i \in V_{\text{train}}} \text{CE}(y_i, y_{\text{final},i}; \boldsymbol{\theta}_2)$ is maximized.

Return $\boldsymbol{Y}_{\text{final}}$

---

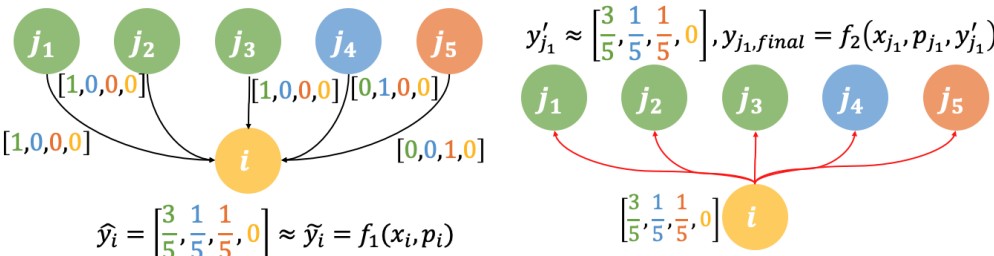

(a) MLaP Forward & Neighbor Model        (b) MLaP Backward & Final Model

Figure 3: Example. For node $i$ we want to learn a model that takes $i$'s features $\boldsymbol{x}_i \in \mathbb{R}^{d_X}$, and PEs $\boldsymbol{p}_i \in \mathbb{R}^{d_P}$ and predict a value $\tilde{\boldsymbol{y}}_i \in \mathbb{R}^c$ that matches the label distribution of it's neighbors neighbors $\hat{\boldsymbol{y}}_i$ using a shallow model. Next, we want to propagate (outside the training loop) the (predicted) distribution of a node back to its neighbors and use it together with the ego features and the PEs to make a prediction about a node's own label. We propagate $\tilde{\boldsymbol{y}}_i$ to its neighbors $j_1$ to $j_5$. For example, for $j_1$, we encode the propagated distribution estimate $\tilde{\boldsymbol{y}}_i$ from $i$ to form $\boldsymbol{y}_{j_1}'$. We predict the label by using $\boldsymbol{y}_{j_1}', \boldsymbol{x}_{j_1}, \boldsymbol{p}_{j_1}$.

In our example in Figure 3(a), we calculate the distribution of node $i$'s neighbors which is $(3/5, 1/5, 1/5, 0)$.

Then, we train a model that predicts the distribution of neighbors, which we denote with $\tilde{\boldsymbol{y}}_i$ using the ego features $\{\boldsymbol{x}_i\}_{i \in V_{\text{train}}}$ and the PEs $\{\boldsymbol{p}_i\}_{i \in V_{\text{train}}}$ and maximize the negative cross-entropy with treating $\{\hat{\boldsymbol{y}}_i\}_{i \in V_{\text{train}}}$ as ground truth labels; namely we maximize

$$\mathcal{L}_{\text{CE, 1}}(\boldsymbol{\theta}_1) = \sum_{i \in V_{\text{train}}} \sum_{l \in [c]} \hat{\boldsymbol{y}}_{i,l} \log(\tilde{\boldsymbol{y}}_{i,l}), \tag{2}$$

where $\tilde{\boldsymbol{y}}_i = f_1(\boldsymbol{x}_i, \boldsymbol{p}_i; \boldsymbol{\theta}_1)$ and $\boldsymbol{\theta}_1 \in \Theta_1$ is a learnable parameter vector. Although in this paper we assume to be in the *transductive setting*, this step allows us to be inductive (see Appendix B). In Section 3.3 we give a theoretical justification of this step, namely *"why is it good to use a parametric model to predict the distribution of neighbors (i.e., a parametric model vs. neighborhood statistics)?"*.

Again, in the example of Figure 3(a), we train a model to learn the distribution of $i$'s neighbors.

Finally, we propagate the predicted soft-labels $\tilde{\boldsymbol{y}}_i$ back to the original nodes, i.e. calculate

$$\boldsymbol{y}_i' = \frac{\sum_{j \in V:(i,j) \in E} \tilde{\boldsymbol{y}}_j}{|\{j \in V : (i,j) \in E\}|} \quad \forall i \in V_{\text{train}}, \tag{3}$$

where the soft labels $\{\tilde{\boldsymbol{y}}_i\}_{i \in V_{\text{train}}}$ have been computed with the parameter $\boldsymbol{\theta}_1^*$ of the epoch with the best validation accuracy from model $f_1(\cdot|\boldsymbol{\theta}_1)$. We call this step MLaP Backward. In the example (Figure 3(b)), this means propagating back the learned distribution from node $i$ – which are close to $(3/5, 1/5, 1/5, 0)$ – back to $i$'s neighbors.

*3rd Stage (Final Model):* We make the final predictions $\boldsymbol{y}_{\text{final, i}} = f_2(\boldsymbol{x}_i, \boldsymbol{p}_i, \boldsymbol{y}_i'; \boldsymbol{\theta}_2)$ by combining the ego embeddings, PEs, and the (back)-propagated soft labels ($\boldsymbol{\theta}_2$ is a learnable parameter vector). We use the soft labels $\tilde{\boldsymbol{y}}_i$ instead of the actual labels one-hot ($y_i$) in order to avoid label leakage, which hurts performance (see also (Shi et al., 2020) for a different way to combat label leakage). Finally, we maximize the negative cross-entropy with respect to a node's own labels,

$$\mathcal{L}_{\text{CE, 2}}(\boldsymbol{\theta}_2) = \sum_{i \in V_{\text{train}}} \sum_{l \in [c]} \mathbb{I}\{y_i = l\} \log(\boldsymbol{y}_{\text{final, i},l}), \tag{4}$$

Finally, in our example in Figure 3(b), this corresponds to using the propagated distribution as one of the inputs in the models we train for each of the nodes $j_1, \ldots, j_5$.

Overall, Stage 2 corresponds to learning the neighbor distributions and propagating these distributions, and Stage 3 uses these distributions to train a new model which predicts a node's labels. In Section 3.3, we prove that such a two-step procedure incurs lower errors than directly using the features to predict a node's labels.

***Complexity:*** GLINKX is highly scalable as it can utilize existing machine learning architecture efficiently since it performs message passing a constant number of times by paying an $O(mc)$ cost, where the dimensionality of classes $c$ is usually small (compared to $d_X$ that GCNs rely on). In both Stages 2 and 3 of Algorithm 1, we train node-level MLPs, which allow us to leverage i.i.d. (row-wise) mini-batching, like tabular data; our complexity is similar to other shallow methods (LINKX, FSGNN) (Lim et al., 2021; Maurya et al., 2021). This, combined with the propagation outside the training loops, circumvents the bottlenecks faced by GNNs. Finally, as with every method, the inference complexity is also a function of how many parameters the model has, which also affects the runtime. For more details, refer Appendix A.1.

***Complementarity:*** Different components of GLINKX provide a *complementary* signal to components proposed in the GNN literature (Maurya et al., 2021; Zhang et al., 2022b; Rossi et al., 2020). One can combine GLINKX with existing architectures (e.g. feature propagations (Maurya et al., 2021; Rossi et al., 2020), label propagations (Zhang et al., 2022b)) for potential metric gains. For example, SIGN computes a series of $r \in \mathbb{N}$ feature propagations $[X, \Phi X, \Phi^2 X, \ldots, \Phi^r X]$ where $\Phi$ is a matrix (e.g., normalized adjacency or normalized Laplacian) as a preprocessing step. We can include this complementary signal, namely, embed each of the propagated features and combine them in the 3rd Stage to GLINKX. Overall, although in this paper we want to keep GLINKX simple to highlight its main components, we conjecture that adding more components to GLINKX would improve its performance on datasets with highly variable homophily (see Figure 1).

***Varying Homophily:*** Graphs with monophily experience homophily, heterophily, or both. For instance, in the yelp-chi dataset – where we classify a review as spam/non-spam (see Figure 1) – we observe a case of monophily together with varying homophily. Specifically in this dataset, spam reviews are linked to non-spam reviews, and non-spam reviews usually connect to other non-spam reviews, which makes the node homophily distribution bimodal. Here the 2nd-order similarity makes the MLaP mechanism effective.

### 3.3 Theoretical Analysis

***Justification of MLaP (Stage 2):*** In the MLaP stage, we train a parametric model to learn the distribution of a node's neighbors from the node features $\boldsymbol{\xi}_i$[6]. Arguably, we can learn such a distribution naïvely by counting the neighbors $i$ that belong to each class. This motivates our first theoretical result. In summary, we show that training a parametric model for learning the distribution of a node's neighbors (as in Stage 2) yields a lower error than the naïve solution. Below we present the Theorem 1 for undirected graphs (the case of directed graphs is the same, but we omit it for simplicity of exposition):

**Theorem 1.** *Let $G([n], E)$ be an undirected graph of minimum degree $K > c^2$ and let $\boldsymbol{Q}_i \in \Gamma_c$ be the distribution of $i$'s neighbors, for every $i \in [n]$. The following two facts are true (under standard assumptions for SGD and the losses):*

1. *Let $\widehat{\boldsymbol{Q}}_i$ be the sample average of $\boldsymbol{Q}_i$, i.e.*

$$\widehat{Q}_{i,j} = \frac{1}{|\mathcal{N}(i)|} \sum_{k \in \mathcal{N}(i)} \mathbb{I}\{y_k = j\}.$$

   *Then, for every $i \in [n]$, we have that*

$$\max_{j \in [c]} \mathbb{E}_{\boldsymbol{Q}_i}[|Q_{i,j} - \widehat{Q}_{i,j}|] \leq \mathbb{E}_{\boldsymbol{Q}_i}[\|\boldsymbol{Q}_i - \widehat{\boldsymbol{Q}}_i\|_\infty] \leq O\left(\sqrt{\frac{\log(Kc)}{K}}\right).$$

2. *Let $q(\cdot|\boldsymbol{\xi}_i; \boldsymbol{\theta})$ be a model parametrized by $\boldsymbol{\theta} \in \mathbb{R}^D$ that uses the features $\boldsymbol{\xi}_i$ of each node $i$ to predict $\boldsymbol{Q}_i$. We estimate the parameter $\boldsymbol{\theta}_I$ by running SGD for $t = n$ steps to maximize $\mathcal{L}(\boldsymbol{\theta}) = \frac{1}{n} \sum_{i=1}^{n} \sum_{j=1}^{c} Q_{i,j} \log q(j|\boldsymbol{\xi}_i; \boldsymbol{\theta})$. Then, for every $i \in [n]$, we have that*

$$\max_{j \in [c]} \mathbb{E}[|q(j; \boldsymbol{\xi}_i; \boldsymbol{\theta}_I) - Q_{i,j}|] \leq O\left(\sqrt{\frac{\log n}{n}}\right).$$

   *The expectation is taken over $\boldsymbol{Q}_i$ and the randomness of the SGD.*

The proof can be found in Appendix F. It is evident here that if the minimum degree $K$ is much smaller than $n$, then the parametric model has lower error than the naïve approach, namely $\tilde{O}(n^{-1/2})$ compared to $\tilde{O}(K^{-1/2})$.

***Justification of MLaP and Final Model Stages (Stages 2 and 3):*** We now provide theoretical foundations for the two-stage approach. Specifically, we argue that a two-stage procedure involving learning the distribution of a node's 2nd-hop neighbor distributions (we assume for simplicity, again, that the graph is undirected) first with a parametric model such as in Theorem 1, and then running a two-phase algorithm to learn a parametric model that predicts a node's label, yields a lower error than naïvely training a shallow parametric model to learn a node's labels. The first phase of the two-phase algorithm involves training the model first by minimizing the cross-entropy between the predictions and the 2nd-hop neighborhood distribution. Then the model trains a joint objective that uses the learned neighbor distributions and the actual labels starting from the model learned in the previous phase.

**Theorem 2.** *Let $G([n], E)$ be an undirected graph of minimum degree $K > c^2$ and, let $\boldsymbol{P}_i$ be the likelihood of node $i$ to be assigned to a different class, and let $\boldsymbol{Q}_i, q(\cdot|\boldsymbol{\xi}_i; \boldsymbol{\theta}_I)$ defined as in Theorem 1. Let $p(\cdot|\boldsymbol{\xi}_i; \boldsymbol{w})$ be a model parametrized by $\boldsymbol{w} \in \mathbb{R}^D$ that is used to predict the class assignments $y_i \sim p(\cdot|\boldsymbol{\xi}_i; \boldsymbol{w})$. Let $\boldsymbol{w}_*$ be the optimal parameter. The following are true (under standard assumptions for SGD and the losses):*

1. *The naïve optimization scheme that runs SGD to maximize $\mathcal{G}(\boldsymbol{w}) = \frac{1}{n} \sum_{i=1}^{n} \sum_{j=1}^{c} P_{i,j} \log p(j|\boldsymbol{\xi}_i; \boldsymbol{w})$ for $n$ steps has error*

$$\mathbb{E}[\mathcal{G}(\boldsymbol{w}_{n+1}) - \mathcal{G}(\boldsymbol{w}_*)] \leq O\left(\frac{\log n}{n}\right),$$

   *where the expectation is taken over $\boldsymbol{P}_i$, and the randomness of the SGD.*

---

[6]In Section 3.1, $\boldsymbol{\xi}_i$s correspond to the augmented features $\boldsymbol{\xi}_i = [\boldsymbol{x}_i; \boldsymbol{p}_i]$

Table 1: Small-scale and medium-scale experimental results. (*) = results from the OGB leaderboard.

| | Homophilous Datasets | | Heterophilous Datasets | | |
| --- | --- | --- | --- | --- | --- |
| | PubMed | ogbn-arxiv | squirrel | yelp-chi | arxiv-year |
| $n$ | 19.7K | 169.3K | 5.2K | 169.3K | 45.9K |
| $m$ | 44.3K | 1.16M | 216.9K | 7.73M | 1.16M |
| Homophily (Lim et al., 2021) | 0.66 | 0.41 | 0.02 | 0.05 | 0.27 |
| $d_X$ | 500 | 128 | 2089 | 32 | 128 |
| $c$ | 27 | 40 | 5 | 2 | 5 |
| GLINKX w/ KGEs | 87.95 $\pm$0.30 | **69.27 $\pm$0.25** | 45.83 $\pm$2.89 | 87.82 $\pm$0.20 | **54.09 $\pm$0.61** |
| GLINKX w/ Adjacency | **88.03 $\pm$0.30** | 69.09 $\pm$0.13 | **69.15 $\pm$1.87** | 89.32 $\pm$0.45 | 53.07 $\pm$0.29 |
| Label Propagation (1-hop) | 83.02 $\pm$0.35 | **69.59 $\pm$0.00** | 32.22 $\pm$1.45 | 85.98 $\pm$0.28 | 43.71 $\pm$0.22 |
| LINKX (from (Lim et al., 2021)) | 87.86 $\pm$0.77 | 67.32 $\pm$0.24 | 61.81 $\pm$1.80 | 85.86 $\pm$0.40 | **56.00 $\pm$1.34** |
| LINKX (our runs) | 87.55 $\pm$0.37 | 63.91 $\pm$0.18 | 61.46 $\pm$1.60 | 88.25 $\pm$0.24 | 53.78 $\pm$0.06 |
| GCN w/ 1 Layer | 86.43 $\pm$0.74 | 50.76 $\pm$0.20 | 26.17 $\pm$2.49 | 85.57 $\pm$0.15 | 44.82 $\pm$0.18 |
| GAT w/ 1 Layer | 86.41 $\pm$0.53 | 54.42 $\pm$0.10 | 30.13 $\pm$1.55 | 86.02 $\pm$1.00 | 45.66 $\pm$0.36 |
| FSGNN w/ 1 Layer | 88.93 $\pm$0.31 | 61.82 $\pm$0.84 | 64.06 $\pm$2.69 | 86.36 $\pm$0.36 | 42.86 $\pm$0.22 |
| Higher-order GCN | 86.29 $\pm$0.46 | 71.18 $\pm$0.27 (*) | 24.81 $\pm$1.70 | 85.60 $\pm$0.15 | 44.58 $\pm$0.28 |
| Higher-order GAT | 86.64 $\pm$0.40 | **73.66 $\pm$0.11** (*) | 27.00 $\pm$1.51 | 85.63 $\pm$0.18 | 45.77 $\pm$0.41 |
| Higher-order FSGNN | **89.37 $\pm$0.49** | 69.26 $\pm$0.36 | 68.04 $\pm$2.19 | 86.33 $\pm$0.30 | 44.89 $\pm$0.29 |
| Label Propagation (2-hop) | 83.44 $\pm$0.35 | 69.78 $\pm$0.00 | 43.41 $\pm$1.44 | 85.95 $\pm$0.26 | 46.30 $\pm$0.27 |
| Label Prop. on $\mathbb{I}[A^2 - A - I \geq 0]$ | 82.14 $\pm$0.33 | 9.87 $\pm$0.00 | 24.43 $\pm$1.18 | 85.68 $\pm$0.32 | 23.08 $\pm$0.13 |

Table 2: Ablation Study. We use the hyperparameters of the best run from Table 1 with KGEs.

| | Ablation Type | Stages | All | Remove ego embeddings | Remove propagation | Remove PEs |
| --- | --- | --- | --- | --- | --- | --- |
| Heterophilous | arxiv-year | All Stages | 54.09 $\pm$0.61 | 53.52 $\pm$0.77 | 50.83 $\pm$0.24 | 39.06 $\pm$0.35 |
| | arxiv-year | 3rd Stage | 54.09 $\pm$0.61 | 53.69 $\pm$0.65 | 50.83 $\pm$0.24 | 49.13 $\pm$1.10 |
| Homophilous | ogbn-arxiv | All Stages | 69.27 $\pm$0.25 | 61.26 $\pm$0.33 | 62.70 $\pm$0.34 | 65.64 $\pm$0.18 |
| | ogbn-arxiv | 3rd Stage | 69.27 $\pm$0.25 | 67.60 $\pm$0.39 | 62.70 $\pm$0.34 | 69.62 $\pm$0.15 |

*2. The two-phase optimization scheme that runs SGD to maximize*

$$\widehat{\mathcal{G}}(\boldsymbol{w}) = \frac{1}{n} \sum_{i=1}^{n} \sum_{j=1}^{c} \left( \frac{1}{|\mathcal{N}(i)|} \sum_{k \in \mathcal{N}(i)} q(j|\boldsymbol{\xi}_k; \boldsymbol{\theta}_I) \right) \log p(j|\boldsymbol{\xi}_i; \boldsymbol{w})$$

*for $n_1$ steps, to estimate a solution $\boldsymbol{w}'$ and then runs SGD on the objective $\lambda \widehat{\mathcal{G}}(\boldsymbol{w}) + (1 - \lambda)\mathcal{G}(\boldsymbol{w})$ for $n$ steps starting from $\boldsymbol{w}'$, achieves error*

$$\mathbb{E}[\mathcal{G}(\boldsymbol{w}_{n+1}) - \mathcal{G}(\boldsymbol{w}_*)] \leq O\left( \frac{\sqrt{\log n \log \log n}}{n} \right).$$

*where the expectation is taken over $\boldsymbol{P}_i$, $\boldsymbol{Q}_i$, and the randomness of the SGD.*

You can find the proof in Appendix F. We observe that the two-phase optimization scheme can reduce the error by a factor of $\sqrt{\log n / \log \log n}$ highlighting the importance of using the distribution of the 2nd-hop neighbors of a node to predict its label and holds regardless of the homophily properties of the graph. Also, note that the above two-phase optimization scheme differs from the description of the method we gave in Algorithm 1. The difference is that the distribution of a node's neighbors is embedded into the model in the case of Algorithm 1, and the distribution of a node's neighbors are embedded into the loss function in Theorem 2 as a regularizer. In Algorithm 1, we chose to incorporate this information in the model because using multiple losses harms scalability and makes training harder in practice. In the same spirit, the conception of GCNs (Kipf & Welling, 2016) replaces explicit regularization with the graph Laplacian with the topology into the model (see also Hamilton et al. (2017); Yang et al. (2016)).

## 4 Experiments & Conclusions

**Small-scale and Medium-scale datasets.** We experiment with homophilous and heterophilous datasets (see Table 1). We train KGEs with Pytorch-Biggraph (Lerer et al., 2019; Yang et al., 2014; Wang et al.,

2017). For homophilous datasets, we compare with vanilla GCN and GAT, FSGNN, and Label Propagation (LP). For a fair comparison, we compare with one-layer GCN/GAT/FSGNN/LP since GLINKX is one-hop. We also compare with higher-order (h.o.) GCN/GAT/FSGNN/LP with 2 and 3 layers. In the heterophilous case, we compare with LINKX [7] because it is scalable and is shown to work better than other baselines (e.g., H2GCN, MixHop, etc.) and with FSGNN. For reasons of completeness, we also report numbers for GCN and GAT (1-layer and h.o.), which are known to underperform in heterophilous settings and suffer from message-passing bottlenecks, as well as LP.

Note that we do not compare GLINKX with other more complex methods because (i) lots of GNN-based methods are not scalable[8], (ii) GLINKX is complementary to them (see Section 3.2), and we can incorporate these designs into GLINKX, (iii) for the heterophilous datasets, GLINKX outperforms or is in agreement with LINKX which is we believe is a strong baseline and outperforms or is in agreement with many recent methods (cf. He et al. (2021); Tang et al. (2019); Dai et al. (2022)) while being substantially more scalable and tested on bigger datasets.

Finally, we use a *ResNet* module to combine our algorithm's components from Stages 2 and 3. Details about the hyperparameters we use are in Appendix C.

In the heterophilous datasets, GLINKX is better/competitive with the baselines. Moreover, the performance gap between using KGEs and adjacency embeddings shrinks as the dataset grows.

In the homophilous datasets, GLINKX outperforms 1-layer GCN/GAT/LP/FSGNN and LINKX. In PubMed, GLINKX beats h.o. GCN/GAT and in arxiv-year GLINKX is very close to the performance of GCN/GAT.

It is important to highlight that the higher-order GNN methods (GCN/GAT) are as good as GLINKX or better **only** in the case where the graph is homophilous, since GCNs/GATs have shown to perform poorly in heterophilous graphs; see Zhu et al. (2020); Lim et al. (2021); Di Giovanni et al. (2022); Jin et al. (2022a); Luan et al. (2021), and the references therein.

Finally, we note that GLINKX *produces consistent results across regime shifts*. In detail, in the heterophilous regime, GLINKX performs on par with LINKX; however, when we shift to the homophilous regime, LINKX's performance drops, whereas GLINKX's performance continues to be high. Similarly, while FSGNN performs similarly to GLINKX on the homophilous datasets, we observe a significant performance drop on the heterophilous datasets (see arxiv-year).

**Scalability Experiment.** To show the scalability of GLINKX, we experiment with the ogbn-products dataset from the OGB benchmark (Hu et al., 2020), which has $n = 2.44M$ nodes and $m = 61.8M$ edges. The features have dimension $d_X = 100$ and our aim is to predict the correct class out of $c = 47$ classes. The graph is categorized as a homophilous graph and has a homophily equal to 0.45.

As shown in Table 3, GLINKX outperforms LINKX by a big margin. Moreover, GLINKX performs much better than the h.o. GCN and FSGNN – albeit the higher number of hops – and has performance comparable to the h.o. GAT. However, at the same time, GCN and GAT require neighbor sampling which raises the training time to days, compared to the simple i.i.d. sampling GLINKX performs, which is able to complete within ours, en par with LINKX.

The ogbn-products dataset took 17.53 seconds on average per epoch to train (for both phases), including the propagation costs. At the same time, the 1-layer GCN took 104.39 seconds per epoch on average to train, and the 1-layer GAT took 107.39 seconds on average per epoch to train. This indicates at least *an 83% reduction in training time* (or equivalently our method is approximately 6× faster).

**Ablation Study.** We ablate each component of Algorithm 1 to see each component's performance contribution. We use the hyperparameters of the best model from Table 1. We perform two types of ablations: (i) we remove each of the components from all stages of the training, and (ii) we remove the corresponding components only from the 3rd Stage. Except for removing the PEs from the 3rd Stage only on ogbn-arxiv, all components

---

[7]We have run GLINKX with hyperparameter space that is a subset of the sweeps reported in (Lim et al., 2021) due to resource constraints. A bigger hyperparameter search would improve our results.

[8]Here we have added a comparison with vanilla GCN and GAT to motivate the architecture of our method and the usefulness of the MLaP layer, and we should note that the vanilla GCN and GAT have the same scalability problems.

Table 3: Results for ogbn-products. (*) = results from the OGB leaderboard. We have omitted the 1-layer GCN/GAT and LP-based methods since GLINKX repeatedly outperforms these by a big margin in the small and medium-scale datasets.

| | |
|---|---|
| GLINKX w/ KGEs | 78.15 ±0.10 |
| LINKX | 69.02 ±0.61 |
| GCN w/ 1 Layer | 66.28 ±0.12 |
| GAT w/ 1 Layer | 65.31 ±0.27 |
| FSGNN w/ 1 Layer | 70.44 ±0.15 |
| Higher-order GCN | 75.64 ±0.21 (*) |
| Higher-order GAT | 79.45 ±0.59 (*) |
| Higher-order FSGNN | 76.03 ±0.33 |

contribute to increased performance on both datasets. Note that adding PEs in the 1st Stage does improve performance, suggesting the primary use case of PEs.

## 5 Conclusion

We present GLINKX, a simple, scalable shallow method for node classification in homophilous and heterophilous graphs that combines three components: (i) ego embeddings, (ii) PEs, and (iii) monophilous propagations. As future work, (i) GLINKX can be extended in heterogeneous graphs, (ii) use more expressive methods such as attention or Wasserstein barycenters (Cuturi & Doucet, 2014) for averaging the low-dimensional messages, and (iii) add complementary signals. While our method is outperformed by the current GNN-based state-of-the-art methods (Rossi et al., 2023; Luan et al., 2021), our *simple* and *scalable* design design avoids the scalability bottlenecks of GNNs which include neighborhood sampling which is a very costly operation, by leveraging propagations outside of the training loop, and can be easily and efficiently deployed in an industrial scale with modern infrastructure. Our extensive evaluation on a database of six datasets with various sizes from the homophilous and heterophilous regime, and the theoretical error bounds we provide, justify our design choices and show that GLINKX is able to perform well and consistently across regime shifts.

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
