# OpenReview forum: "A Simple Unified Method for Node Classification on Homophilous and Heterophilous Graphs"
_TMLR — Rejected by TMLR_

### Review · Reviewer_b8oQ · 2023-07-17

**Summary Of Contributions:**

This paper proposes GLINKX, a method that works for both homophilous and heterophilous node classification, and is also scalable. The proposed GLINKX leverages a novel notion called monophilous (i.e. the neighbors of a node tend to be of the same class). To leverage the property, the authors propose a two-stage bilevel label propagation method to propagate the 2nd-order neighbor label distribution back to the original node. The authors also design a positional encoding scheme based on knowledge graph embedding (KGE). Experiments show that the proposed method is competitive in terms of both homophilous and heterophilous graphs, and is also scalable compared to GNN methods.

**Audience:**

Yes

**Broader Impact Concerns:**

No.

**Claims And Evidence:**

No

**Requested Changes:**

Please see 'Weaknesses' and try to address them with revisions or clarifications.

**Strengths And Weaknesses:**

## Strengths
- The paper aims to unify homophilous and heterophilous node classification with a unified method, which is a novel and interesting attempt as existing works may primarily tackle one of them.
- The introduced notion of monophily seems interesting and has the potential for unifying homophily and heterophily.
- The proposed method is competitive in both homophilous and heterophilous classification and is also scalable.
- Experiments are all done on relatively large datasets.

## Weaknesses
- Although monophily is conceptually appealing to unify the homophily and heterophily graphs, there is no quantitative analysis on how monophily holds true on the real-world graphs. It would be better if the authors can use some quantitative metrics to verify the monophily property.
- It is unclear why the authors specifically choose knowledge graph embeddings for learning positional encodings. Specifically, there are many methods for ordinary graph embedding (i.e. not knowledge graphs), such as DeepWalk (KDD 2014), Node2vec (KDD 2016), etc. Moreover, they are also implicitly factorizing the adjacency matrix (Levy and Goldberg 2014). Therefore, it is unclear why the authors specifically use KGE (namely also embedding the relation instead of nodes). More justifications should be needed.
- Are the experiments in Table 1 and Table 3 in the transductive or inductive setting? The authors say on Page 2 that the paper focuses on the transductive setting, but this does not seem convincing to demonstrate the contribution of GLINKX. First, the authors say in Section 1 that LINKX is not practical because it is not inductive, and failing to show GLINKX in the inductive setting means that a key advantage of GLINKX is not demonstrated. Moreover, GNN baselines are often evaluated with inductive settings. It is unclear whether the comparison is fair.
- The authors use the subtitle 'scalability experiment' in Section 4, but the authors fail to show the training time of GLINKX, which does not seem convincing as a 'scalability experiment'.
- It seems that the proposed method GLINKX relies heavily on ground truth labels (to obtain the label distribution). Therefore, GLINKX may be sensitive to the number of labels. It is good to have a sensitivity analysis of GLINKX (as well as other semi-supervised GNN baselines).
- It seems that the proposed notion of monophily is similar to the 'second-order similarity' proposed in LINE (Tang et al. 2015). The authors are suggested to justify the similarity and difference.
- (Minor) The arrangement of  figures and algorithms makes the paper hard to read. For example, Figure 3 is referenced on Page 2 but lies on 9. Figure 2 is referenced on Page 4 but lies on Page 8. Algorithm 1 is referenced on Page 6 but given on Page 3. Please consider revising the organization.

(Levy and Goldberg 2014) Neural Word Embedding as Implicit Matrix Factorization, NIPS 2014.
(Tang et al. 2015) LINE: Large-scale Information Network Embedding, WWW 2015.

---

### Review · Reviewer_XhwM · 2023-07-28

**Summary Of Contributions:**

This paper considers the node classification task on one graph in a supervised setting. The authors propose a new algorithm based on local computations that can scale to large graphs.

**Audience:**

No

**Claims And Evidence:**

No

**Requested Changes:**

As explained above, the paper is not ready for publication. I am not sure if the proofs can be repaired. The experimental section is weak.
The organization of the paper is also problematic. In particular, there are a lot of places in the Introduction where some notions are defined only much later in the paper. As a result, the paragraph at the beginning of page 2 cannot be understood with the context provided. I am not sure the authors should introduce knowledge graphs as they are actually using them only with one type of relation.

**Strengths And Weaknesses:**

There are some main issues to be addressed before any possible publication:

1- The theoretical analysis does not make any sense. In Theorem 1, the authors introduce a likelihood but never introduce a probabilistic model. I do not understand at all what they mean here. What is Q_i? and what is the measure of probability used for the expectation? Looking at the proof does not help. In the proof, there is a new notation P_k, that has never been introduced. Why can you assume that the loss is a smooth function of the parameters? In the proof, what do you mean by: "Let \theta_1 = \theta_{n+1}" (top of page 23)?

2- I do not understand why in stage 2 of the algorithm, you propagate the predicted soft-labels instead of propagating the true \hat{y} given in equation (1).

3- In the experiments, details need to be provided. The measure of performance is never defined. The main motivation for the work is to derive a scalable algorithm. In the introduction, the authors say that GNNs do not scale but Table 3 provides performances for GNN on large datasets. To make their points, the authors should provide numerical evidence that their algorithm is faster than those GNNs.

---

> ### Author Response · Authors · 2023-08-13
> **Propagating Labels, Theory, Scalability (Part 1/2)**
>
> We would like to thank the Reviewer for the insightful comments and valuable feedback. We have incorporated the appropriate changes in our uploaded revision. In the sequel, we address the indicated weaknesses:
>
> ### Why don’t we propagate labels?
>
> There are several reasons why we do not choose to propagate the actual labels in GLINKX, instead, we choose to train a model to learn the distribution of neighbors.
>
> Firstly, this technique enables our method to be inductive, whereas if we used the actual labels, our method would not be inductive.
>
> Secondly, using a model instead of the actual labels can enable our model to be trained in an environment with a sparsity of labels and also helps in avoiding overfitting.
>
> Finally, we prove in Theorem 2 that if we rely in this two-step procedure, we achieve a better error bound by a factor of $\sqrt {\log n / \log \log n}$ compared to training a simple MLP model.
>
> ### Theory
>
> In our revision, we have revised the statements of the theorem, the notation, and made all the corresponding changes in the proofs.
>
> **Theorem 1:**
> * In Theorem 1, $Q_i$ corresponds to the true probability distribution of the labels of the neighbors of node $i$. Theorem 1 basically states we can either estimate $Q_i$ by just taking a sample average $\hat Q_i$ of the labels of the neighbors of node $i$, or by training a parametric model $q_i( \cdot | \theta)$ with SGD. We show that under reasonable conditions for SGD (smoothness, PL condition), training the second model would give us a lower error in estimating $Q_i$.
> * The expectations are taken over $Q_i$ for the first case and over $Q_i$  and the randomness of SGD for the second case.
> * Therefore, Theorem 1 argues that to learn the distributions of the neighbors (see our response to the previous question for the benefits this idea has, which include inductivity, tackling label sparsity, and better behavior towards overfitting), a parametric model which is what we use in Stage 2 is actually useful.
>
> **Theorem 2**
> * In Theorem 2, $P_i$ is the true distribution of a node's labels $i$.
> * Regarding the SGD assumptions, we have followed the standard assumptions for SGD in the convex optimization literature.
> * Thank you for pointing this out regarding the notation; we understand that this may have caused confusion. We meant that $\theta_1$ is the parameter vector after SGD ends (which is what we have in the statement of Theorem 1 and then Theorem 2). In our uploaded revision, we corrected the notation to $\theta_I$ to indicate the parameter vector after _Phase I_ ends.
>
> ### Scalability and Measures of Performance
>
> **See also our [general comment](https://openreview.net/forum?id=7JKFHoXYEG&noteId=u0Ew4UInvz) regarding the runtime of our method on ogbn-products where our method runs 6x faster than GCN/GAT**
>
> In order to run a message-passing GNN in a large-scale network such as a social network, one must use neighborhood sampling during training, which is a very costly process, in terms of both time, and memory, because directly running GNNs is impossible to fit in GPU memory, even for moderately-sized graphs (~100K nodes). These problems have been well-documented in the GNN literature, for example, see the following references:
>
> * Frasca, Fabrizio, et al. "Sign: Scalable inception graph neural networks." ICML 2020 Workshop on Graph Representation Learning and Beyond
> * Frasca, Fabrizio, Bronstein, Michael. “Simple scalable graph neural networks”. Twitter engineering blog. Accessed at Aug 5, 2023. (The references inside this blog post are also useful pointers outlining why GNNs are impractical to scale in large graphs).
> * Lim, Derek, et al. "Large scale learning on non-homophilous graphs: New benchmarks and strong simple methods." Advances in Neural Information Processing Systems 34 (2021): 20887-20902.
> * Bojchevski, Aleksandar, et al. "Scaling graph neural networks with approximate pagerank." Proceedings of the 26th ACM SIGKDD International Conference on Knowledge Discovery & Data Mining. 2020.
>  * Ying, Rex, et al. "Graph convolutional neural networks for web-scale recommender systems." Proceedings of the 24th ACM SIGKDD international conference on knowledge discovery & data mining. 2018.
>
> Moreover, regarding GNNs and sampling, you can always fix the size of the sampling you do for GNNs, but you need to drop a lot of edges which will impact the performance of the GNN since it costs $O(|E|)$ every training epoch anyways.
>
> On the contrary, our method works the same way as simple MLPs and does not require propagations during the training phase, which avoids the neighborhood sampling problem, and significantly decreases the training time. First-order propagations outside training are easy even on an industrial scale with a simple join and then a group by operation which can be implemented easily and very efficiently in Hadoop, MapReduce, etc.

---

> ### Author Response · Authors · 2023-08-13
> **KGEs, Introduction (Part 2/2)**
>
> Due to OpenReview's 5000 character limit in the responses we could not fit our response in one answer. The remainder of our response is below:
>
> ### Knowledge Graph Embeddings
>
> Knowledge graph embeddings (KGEs) can definitely be used in simple graphs, because simple graphs are special cases of heterogeneous graphs with one relation. Our paper uses KGEs to produce node embeddings for several reasons, which we have also outlined in our response to Reviewer b8oQ (link).
>
> ### Introduction
>
> We have significantly revised the introduction to make sure that everything is adequately explained, and our contributions are clearly laid out.

---

### Review · Reviewer_g3NA · 2023-08-02

**Summary Of Contributions:**

This work proposes a new graph deep learning approach to improve the node classification performance over both homophilic and heterophilic graphs. The key idea is to leverage the property that the labels of nodes that share a common neighbor tend to be monophilous.  The algorithm first trains a model based on single-node features to predict the aggregated labels of neighbors. Then, use the learned model to predict the unlabeled nodes and propagate the obtained pseudo labels to neighbors. A second model uses the received aggregated pseudo labels combined with node-self features to make the final prediction.  The work also finds that node positional embeddings derived from knowledge graph embedding algorithms are effective. Overall, the algorithm achieves good prediction accuracy.

**Audience:**

Yes

**Claims And Evidence:**

Yes

**Requested Changes:**

I expect that the authors can find more datasets that can show the effectiveness of the proposed approach. Also, add discussion on [3].


**Strengths And Weaknesses:**

Strengths:
1. The overall algorithm of this work mainly combines two previous frameworks: LINKX [1] (combining with node positional embeddings without keeping permutation equivariant) and C&S [2] (label propagation and combine it with NNs), but the combination is not trivial to me. It is quite interesting to see such a kind of combination work.

2. The idea of leveraging  monophilous labels of the neighbors of a node is also new.

3. The empirical performance of the model is good.

4. Overall the paper is written well, and it is easy to follow.

5. The work also provides some theory to explain why the label propagation may help.

[1] Large Scale Learning on Non-Homophilous Graphs: New Benchmarks and Strong Simple Methods, Lim et al., NeurIPS 2021
[2] Combining label propagation and simple models out-performs graph neural networks, Huang et al., ICLR 2021

Weaknesses:
1. The performance of the model is not uniformly the best. In particular, over homophilic graphs, there are some non-trivial gaps between the proposed approach and the state-of-the-art method. Over some heterophilic graphs, say arxiv-year, the performance of the proposed method is not the best as well.

2. In most cases, using node positional encoding achieves even worse performance than using rows in the adjacency matrix (LINKX adopts).

3. Miss the discussion on the key reference [3] that also uses generalized network embedding methods as positional encoding for node-level tasks.

[3] Equivariant and Stable Positional Encoding for More Powerful Graph Neural Networks, Wang et al. ICLR 2022

---

> ### Author Response · Authors · 2023-08-13
> **Discussion of Wang et al. (2022) and Experiments**
>
> We would like to thank the Reviewer for the constructive comments and valuable feedback. We have addressed the concerns in the responses below and have uploaded a revised manuscript.
>
> ### Discussion of Wang et al. (2022)
>
> Thanks for bringing this work to our attention. This is an excellent paper; we have incorporated the discussion in the revision (see our additions in red color in Section 3.1 of the revised manuscript).
>
> ### Experiments
>
> **(1)** Our method aims to deliver a _simple_ and _scalable_ method that can work in both the homophilous and heterophilous settings, which avoids what GNNs do. As such, we do not claim that our method is state-of-the-art but that it is a simple and scalable method that can work on an industrial scale, which is something that may not be possible/or trivial to do with GNNs because of the complex infrastructure that is needed.
>
> For example, for the ogbn-products dataset, which is a very reasonably-sized dataset, [our method is ~6x faster](https://openreview.net/forum?id=7JKFHoXYEG&noteId=u0Ew4UInvz) (or achieves an 83% reduction in training time) than GCN/GAT. We have included this in our revised manuscript.
>
> The state-of-the-art specialized methods on the homophilous datasets usually do not work on heterophilous graphs and vice-versa. Moreover, we know that some methods perform well across the spectrum, such as ACM and DirGNN. However, methods similar to ACM or DirGNN are architecturally similar to GCNs and suffer from scalability problems, so they are not appropriate baselines to compare with.
>
> Finally, we believe that we showed the performance of our method in a variety of datasets and with a range of homophily. In addition to that, we have provided several theoretical results that show the advantages of our design choices.
>
>  * ACM: Luan, Sitao, et al. "Is heterophily a real nightmare for graph neural networks to do node classification?." arXiv preprint arXiv:2109.05641 (2021).
>  * DirGNN: Rossi, Emanuele, et al. "Edge Directionality Improves Learning on Heterophilic Graphs." arXiv preprint arXiv:2305.10498 (2023).
>
> **(2)** We agree that sometimes the KGEs deteriorate the accuracy. However, we believe that the loss is minimal compared to the scalability and flexibility benefit we gain (see our responses to the two other reviewers) by using KGEs which can essentially compress the adjacency representation to ~ 400 dimensions, compared to the high number of dimensions that an exact embedding would have which could be of the order of at least millions of dimensions for most of industry datasets, see e.g.
>
> *  El-Kishky, Ahmed, et al. "Twhin: Embedding the twitter heterogeneous information network for personalized recommendation." Proceedings of the 28th ACM SIGKDD conference on knowledge discovery and data mining. 2022.

---

### Author Response · Authors · 2023-08-13
**Scalability of our Method, Runtime, and Performance on ogbn-products**

### Scalability of our Method, Runtime, and Performance on ogbn-products

**We have added the runtime on ogbn-products together with additional runtime comparison with GCN and GAT for our method to show our method’s _scalability_.**

Our method ogbn-products was 17.53 seconds on average per epoch to train for both phases, including the propagation costs. At the same time, a 1-layer GCN with 16-dim hidden channels took 104.39 sec per epoch to train on average, and a 1-layer GAT with two attention heads and 16-dim hidden channels took 107.39 per epoch to train on average. Both numbers indicate an >6x speedup (>83% reduction in training time).

---

### Decision · Action_Editors · 2023-09-17

**Recommendation:** Reject

**Comment:**

Concretely, the proposed MLaP method first calculates the averaged label distribution for each node from the labels of its neighbors, \hat{y}, then trains a parametric model to predict this $\hat{y}$.  The analysis claims that the prediction made by the parametric model is more accurate compared to $\hat{y}$ itself in terms of approximating the true label distribution.  Furthermore, the analysis claims that the approximation error of the parametric model goes down as the number of SGD steps, n, increases.

This analysis is wrong even just looking at the claims themselves.  When optimizing a model to approximate $\hat{y}$, even assuming perfect approximation, the model would at best do as good as the target $\hat{y}$.  The approximation error, therefore, should not be better than achievable by $\hat{y}$.  If the claim that the approximation error decreases with SGD steps n were true, then as $n \rightarrow \infty$ we would have 0 approximation error, which is obviously wrong.  The authors should probably check if all the simplification and independence assumptions hold in their setup.  I also found that the proofs and claims made in the analysis didn’t state the setting and assumptions very clearly, which should be improved.

In my opinion the good thing about learning a parametric model to predict the smoothed label distribution is that the parametric model can then be used for nodes that e.g. does not have labeled neighbors and can therefore generalize, and can also potentially be used in the inductive setting, that’s not possible by a pure label propagation approach.

In the ablation experiments, the authors also did not show that this learned parametric model works better than just the empirical average (or maybe that’s not possible).  It would be good to demonstrate this somehow.

**Audience:**

The paper proposes a scalable model for graph data.  The community working on graph data and graph neural networks would be interested in it.

The 2-hop MLaP method is somewhat new and some people might find it interesting.

**Claims And Evidence:**

This paper proposes GLINKX, a model for graph data that is claimed to be simple, scalable and effective on both homophilous and heterophilous graphs.  The ingredients are mostly 1) using knowledge graph embeddings as position embeddings for the nodes and 2) a novel 2-hop label propagation approach, that first learns a parametric model to predict the averaged label distribution averaged from a node’s neighbors, and then propagates the predicted smoothed distribution back to each node from the neighbors and use that as additional features to do the final prediction of a node’s labels.

In terms of claims:
* The method is simple and scalable: the ideas are all easy to understand, and the approach is scalable as all the training can be done at the individual node level and therefore can benefit from the scalable i.i.d. training without worrying about the graph structures.
* The method seems to be effective on homophilous and heterophilous graphs as shown by the experiments, the proposed method is not the best on all tasks but is mostly competitive with the selected baselines.

The paper makes the additional theoretical analysis about the proposed 2-hop label propagation mechanism in Section 3.3 that their proposed MLaP method is more accurate than taking the sample average of the labels from neighbors, which constitute a significant part of this paper.  This analysis, however, seems flawed (see detailed comments).

**Resubmission Of Major Revision:**

The authors may consider submitting a major revision at a later time.